# Bacterial but Not Fungal Rhizosphere Community Composition Differ among Perennial Grass Ecotypes under Abiotic Environmental Stress

Soumyadev Sarkar,[a] Abigail Kamke,[a] Kaitlyn Ward,[a] Aoesta K. Rudick,[d] Sara G. Baer,[d,e] QingHong Ran,[a] Brandi Feehan,[a] Shiva Thapa,[b] Lauren Anderson,[a] Matthew Galliart,[c] Ari Jumpponen,[a] Loretta Johnson,[a] ⓘ Sonny T. M. Lee[a]

[a]Division of Biology, Kansas State University, Manhattan, Kansas, USA
[b]Department of Biology, University of North Carolina at Greensboro, Greensboro, North Carolina, USA
[c]Department of Biological Sciences, Fort Hays State University, Hays, Kansas, USA
[d]Kansas Biological Survey & Center for Ecological Research, University of Kansas, Lawrence, Kansas, USA
[e]Department of Ecology and Evolutionary Biology, University of Kansas, Lawrence, Kansas, USA

Abigail Kamke and Kaitlyn Ward contributed equally to this article. The author order was based on their last names alphabetically.

**ABSTRACT** Environmental change, especially frequent droughts, is predicted to detrimentally impact the North American perennial grasslands. Consistent dry spells will affect plant communities as well as their associated rhizobiomes, possibly altering the plant host performance under environmental stress. Therefore, there is a need to understand the impact of drought on the rhizobiome, and how the rhizobiome may modulate host performance and ameliorate its response to drought stress. In this study, we analyzed bacterial and fungal communities in the rhizospheres of three ecotypes (dry, mesic, and wet) of dominant prairie grass, *Andropogon gerardii*. The ecotypes were established in 2010 in a common garden design and grown for a decade under persistent dry conditions at the arid margin of the species' range in Colby, Kansas. The experiment aimed to answer whether and to what extent do the different ecotypes maintain or recruit distinct rhizobiomes after 10 years in an arid climate. In order to answer this question, we screened the bacterial and fungal rhizobiome profiles of the ecotypes under the arid conditions of western Kansas as a surrogate for future climate environmental stress using 16S rRNA and ITS2 metabarcoding sequencing. Under these conditions, bacterial communities differed compositionally among the *A. gerardii* ecotypes, whereas the fungal communities did not. The ecotypes were instrumental in driving the differences among bacterial rhizobiomes, as the ecotypes maintained distinct bacterial rhizobiomes even after 10 years at the edge of the host species range. This study will aid us to optimize plant productivity through the use of different ecotypes under future abiotic environmental stress, especially drought.

**IMPORTANCE** In this study, we used a 10-year long reciprocal garden system, and reports that different ecotypes (dry, mesic, and wet) of dominant prairie grass, *Andropogon gerardii* can maintain or recruit distinct bacterial but not fungal rhizobiomes after 10 years in an arid environment. We used both 16S rRNA and ITS2 amplicons to analyze the bacterial and fungal communities in the rhizospheres of the respective ecotypes. We showed that *A. gerardii* might regulate the bacterial community to adapt to the arid environment, in which some ecotypes were not adapted to. Our study also suggested a possible tradeoff between the generalist and the specialist bacterial communities in specific environments, which could benefit the plant host. Our study will provide insights into the plant host regulation of the rhizosphere bacterial and fungal communities, especially during frequent drought conditions anticipated in the future.

**KEYWORDS** rhizobiome, bacteria, fungi, drought, ecotypes, grass

Address correspondence to Sonny T. M. Lee, leet1@ksu.edu.

The authors declare no conflict of interest.

The rhizosphere, which plays a pivotal role in plant function by facilitating elemental and water cycling, and uptake of nutrients (1), is densely populated by diverse microbial communities – the rhizobiome. A wide range of complex interactions ranging from symbiotic to competitive among the rhizobiome microorganisms governs the carbon, nitrogen, and phosphorus uptake and transformations (2, 3). These interactions exist not only among the microorganisms but also between the plant hosts, and their associated rhizobiome. For example, plant hosts may selectively attract and/or repel specific soil microbial communities through their root exudates (4, 5). Furthermore, these microorganisms may establish symbiotic relationships with the host plants, safeguarding the host against pathogens (6, 7). These interactions between the plant host and its rhizobiome are likely to be highly specific and ultimately important for community stability, ecosystem functioning, and maintaining soil biodiversity (8).

Global change can have adverse effects on microbe-microbe and plant-microbe interactions, which in turn, can impact the ecology of the rhizosphere and ecosystem function (2). Some studies have focused on the impacts of climate change on rhizospheres (9–12). However, more concerted efforts are needed to fill the knowledge gaps between how rhizobiomes may be influenced by the interactive effects of the plant host and the ever-changing climate. Ultimately, this will help us to maximize plant growth and survival in stressful environments (13–15), especially impacts from drought.

Microorganisms in the rhizosphere are sensitive to environmental conditions (16) and can be good indicators of soil quality (17). Dissecting the rhizosphere bacterial and fungal communities, their functional roles, and their interactions with the plant hosts are crucial to developing future methods to improve drought tolerance and plant productivity. The root-associated microbes can elevate the drought tolerance mechanisms in plants through the physiological and biochemical pathways in the plants (18). Plant-microbe interactions in the rhizosphere have been shown to enhance host resistance to environmental stress and support plant growth. Some of these plant-associated microbial mechanisms include biofilm formation (19), osmotic adjustments (20), changes in phytohormonal levels (18), increase in antioxidant enzymes (18), increase in nutrient and water uptake (21, 22), and optimization of gas exchanges (21, 23). Climate change, characterized by rising temperature and shifted precipitation patterns, has caused the increase in drought frequency (24, 25) and severity (26, 27). Thus, comprehending the extent of the variability among the dominant species in an ecosystem, and how this variability interacts under the predicted arid conditions is important (28). This is especially critical because the dominant species greatly impact ecosystem processes such as carbon assimilation, nutrient cycling, etc. (29). The motivation to understand plant host-microbe interaction in dominant grasses is therefore extremely crucial in the case of tallgrass prairies, in which grasses are responsible for the majority of the carbon fixing, nutrient cycling and biomass (30, 31).

Our studies focus on Big Bluestem, *Andropogon gerardii*, the dominant native grass species in the tallgrass prairies of central North America (32). *A. gerardii* is widely distributed across the Midwest and Northeastern USA (USDA database). Our study focuses on the Central grasslands where this grass dominates, stretching from western Kansas to southern Illinois (33). This precipitation gradient includes a semiarid environment, a region of intermediate precipitation, and a region of heavy rainfall (34). Galliart and colleagues have demonstrated that within this steep precipitation gradient, three genetically distinct *A. gerardii* regional climate ecotypes (dry, mesic, and wet) exist (28, 35), in terms of leaf area, height, and blade width (36), allocation to roots (37), and chlorophyll abundance (38). Climate change models have also predicted a strong phenotypic cline in *A. gerardii* across this longitudinal precipitation gradient (28, 36, 39). However, we have limited understanding on how the three ecotypes and their rhizobiomes are differently affected by a semiarid environment to which some ecotypes may be more adapted to than others. As such, understanding the shifts and interaction between the *A. gerardii* plant host and its associated microbiome at Colby Kansas will provide

insights into exploring the impact of future climate change on grasslands and the microbiome.

The goal of this study was to analyze the composition of bacterial and fungal communities in the rhizobiomes of the dry, mesic, and wet *A. gerardii* ecotypes originating in Hays Kansas (rainfall ~500 mm/year), Manhattan Kansas (rainfall ~870 mm/year) and Carbondale Illinois (rainfall ~1,200 mm/year), respectively. All ecotypes were planted in Colby Kansas (rainfall ~500 mm/year) and grown for 10 years prior to sampling. We asked to what extent do ecotypes of a dominant prairie grass maintain or recruit distinct rhizobiomes after 10 years of growth in a semi-arid climate where precipitation is lower than where the ecotypes originated. We were specifically interested in deciphering the extent of ecotypic variation and/or pressure of a semiarid environment on the dominant tall-grass prairie grass rhizobiome. We postulated that the plant host would exert their ecotypic influences on the rhizobiome even under environmental stress, and thus would observe differences in the ecotypic microbial community. We hypothesized that: (1) because the taxonomic traits are driven by the recruitment of the plant host (40–45), we would be able to identify a core rhizobiome that was associated with the different ecotypes; and (2) because of the semi-arid environment of Colby, we would identify microbial populations which might be more resilient to environmental abiotic stress. This study aims to provide insights necessary to preserve the prairie ecosystems under climate change pressures.

## RESULTS AND DISCUSSION

We dissected the rhizosphere bacterial and fungal communities associated with the dominant tallgrass prairie species, *A. gerardii*. We recovered an average of 17,181 ± 4,935 counts per sample for bacteria, and 37,015 ± 7,394 counts for fungi after primer trimming (Table S1 in the supplemental material). Of the recovered counts, an average of 71.45% bacterial counts and 61.55% fungal counts were annotated to the species level on SILVA and UNITE respectively. Any unknown or unclassified amplicon sequence variants (ASVs) were removed from downstream analyses.

**No differences in bacterial or fungal $\alpha$-diversity among host ecotypes.** We performed the Kruskal-Wallis statistical analyses and found no support for differences in bacterial $\alpha$-diversity ($S_{Obs}$, Shannon's H'index: H = 6.374, $P$ = 0.041, or Faith's PD index: H = 3.626, $P$ = 0.163 and observed ASVs index: H = 3.959, $P$ = 0.138) among *A. gerardii* ecotypes (Fig. 1A). Venn diagrams of the shared bacterial and archaeal ASVs reveal 1,703 shared ASVs (98.66%) among the three ecotypic rhizobiomes (Fig. 1A). These shared ASVs were Actinobacteria, Proteobacteria, Acidobacteria, Verrucomicrobia, Bacteroidetes, Thaumarchaeota, Chloroflexi, Firmicutes, Patescibacteria, Planctomycetes, Armatimonadetes, Gemmatimonadetes, Latescibacteria, Cyanobacteria, Rokubacteria, Entotheonellaeota, Nitrospirae, BRC1, Chlamydiae, Dependentiae, FBP, Elusimicrobia, Deinococcus-Thermus, Fibrobacteres, and WS2. Similar to the bacterial $\alpha$-diversity, we observed no support for differences in the fungal rhizobiome $\alpha$-diversity among the three *A. gerardii* ecotypes when we performed the Kruskal-Wallis statistical analysis ($S_{Obs}$, Shannon's H'index: H = 3.759, $P$ = 0.153, or Faith's PD index: H = 4.798, $P$ = 0.091 and observed ASVs index: H = 3.393, $P$ = 0.183) (Fig. 1B). There was one unique fungal ASV belonging to the wet ecotypes, and none in the other ecotypes. There were 829 (99.28%) overlapping ASVs among the three ecotypes (Fig. 1B). The rhizobiome ASVs that were shared among the dry, wet and mesic ecotypes belonged to Basidiomycota, Ascomycota, Mortierellomycota, Glomeromycota, Kickxellomycota, Chrytridiomycota, Rozellomycota, Aphelidiomycota, and Entomophthoromycota.

**Bacterial composition differed among host ecotypic rhizobiome.** We performed PERMANOVA statistical analyses and showed that bacterial composition at the phylum level differed among the three *A. gerardii* ecotypes (PERMANOVA: Pseudo-F = 4.1963, p [permutation(perm)] = 0.001, p[Monte Carlo(MC)] = 0.002, NMDS; stress = 0.13). We also observed a difference in bacterial composition among the ecotypes at the genus level (PERMANOVA: Pseudo-F = 2.1014, p(perm) = 0.001, p(MC) = 0.003). We analyzed the samples based on Bray-Curtis similarity, and observed that the mesic data cloud

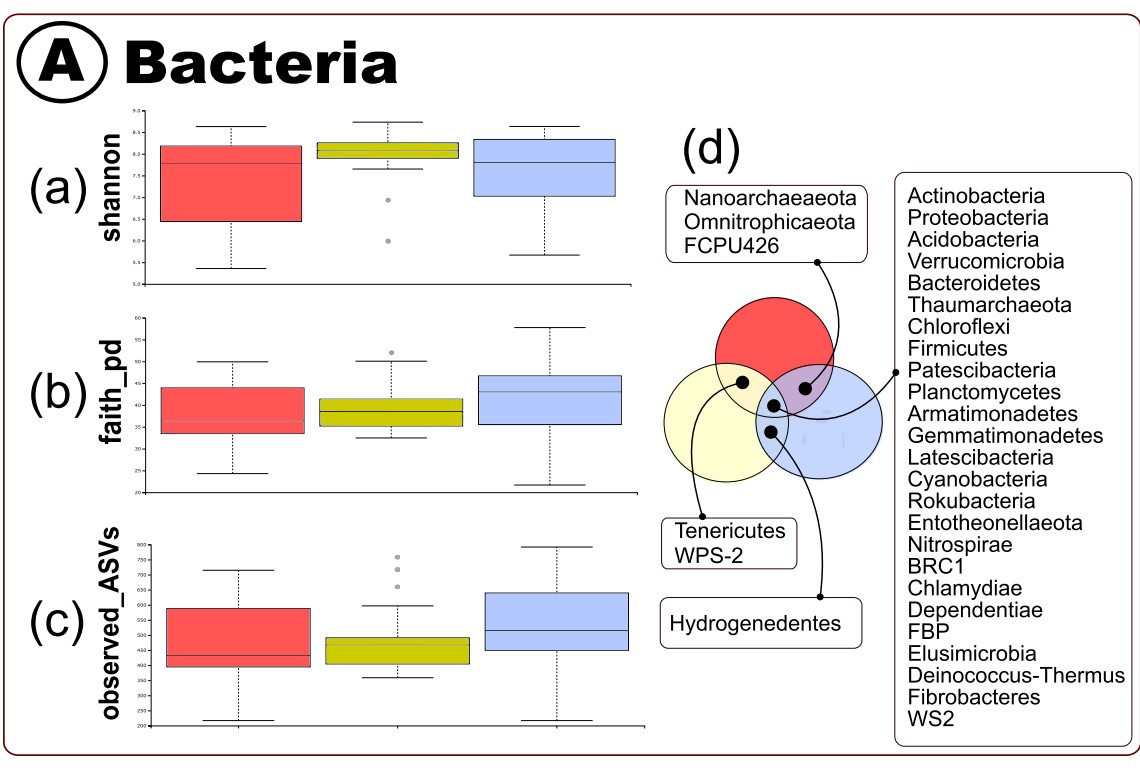

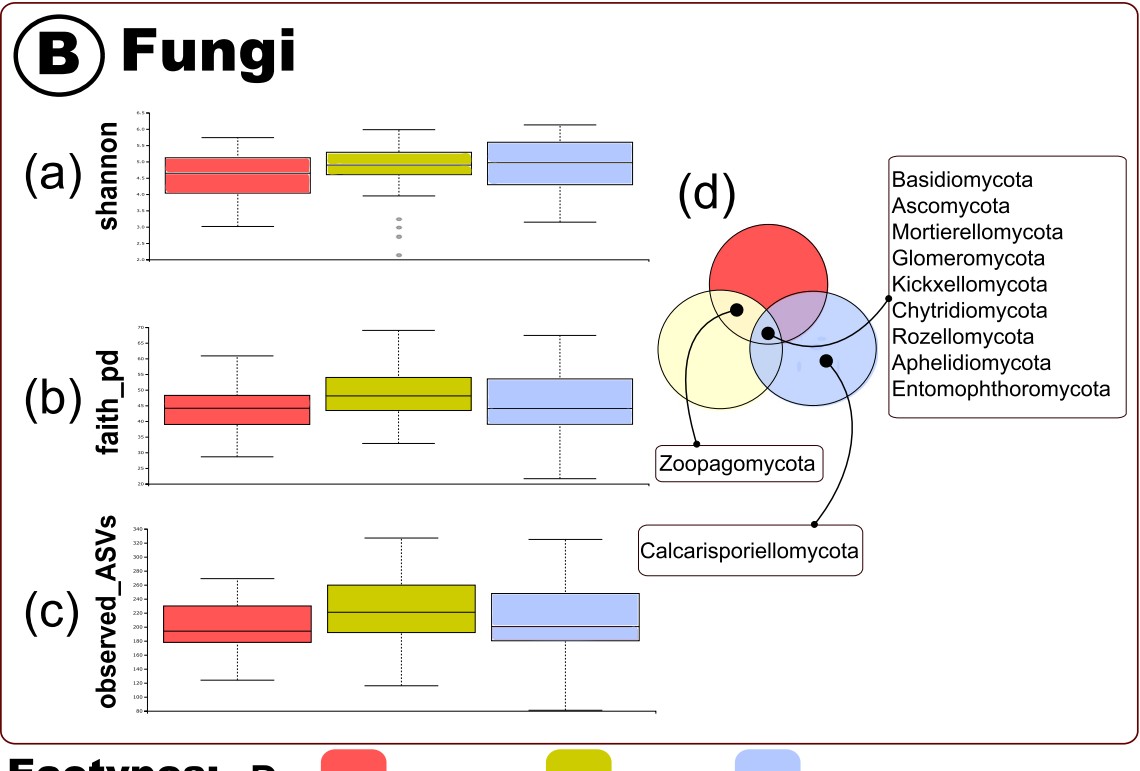

**Ecotypes:** Dry ▮  Mesic ▮  Wet ▮

**FIG 1** A: Bacterial $\alpha$-*Diversity* indices among the dry, mesic, and wet ecotypes. (a) Shannon index, (b) faith-pd index, and (c) observed ASVs index, and (d) Venn diagrams represent the overlapping bacterial and archaeal ASVs among the dry, mesic, and wet rhizobiomes. The bacterial $\alpha$-*Diversity* was not significantly different among the samples. B: Fungal $\alpha$-*Diversity* indices among the dry, mesic, and wet ecotypes. (a) Shannon index (b) faith-pd index and (c) observed ASVs index (d) Venn diagrams representing the overlapping fungal ASVs among the dry, mesic, and wet rhizobiomes. The fungal $\alpha$-*Diversity* was not significantly different among the samples.

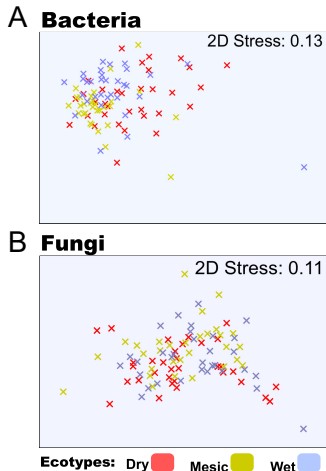

**FIG 2** Bacterial and fungal composition between the dry, mesic, and wet ecotypes. NMDS ordinations were obtained from Bray-Curtis similarity matrix. The matrix was calculated from square-root transformed relative abundance of 16S and ITS 2 rRNA amplicon sequences (A) bacterial community compositions are separated between the three ecotypes (B) fungal community compositions are not separated between the three ecotypes.

dispersion was smaller than that of the dry and wet ecotypes (Fig. 2A). In contrast to bacteria, we did not observe any evidence for differences in the fungal rhizobiome composition among the dry, wet and mesic ecotypic rhizobiomes when we performed the PERMANOVA statistical analyses (PERMANOVA: Pseudo-F = 2.0827, p(perm) = 0.071, p(MC) = 0.08, NMDS; stress = 0.11; Fig. 2B).

**Significant differences in ecotypic bacterial composition.** Based on pairwise Kruskal-Wallis test, bacterial rhizobiome communities were distinct among all three ecotypes (dry v wet: p(MC) = 0.034, dry v mesic: p(MC) = 0.001, wet v mesic: p(MC) = 0.005). The top seven bacterial phyla present in all the three ecotypes are Proteobacteria, Actinobacteria, Acidobacteria, Chloroflexi, Bacteroidetes, Verrucomicrobia, Planctomycetes, and one archaeal taxa, Thaumarchaeota (Fig. 3, Fig. S1 in the supplemental material). Acidobacteria, Bacteroidetes, and Proteobacteria are ubiquitous in soil, suggesting their importance in our samples (46). Similarly, Chloroflexi, Verrucomicrobia, Planctomycetes, and Thaumarchaeota are common soil dwellers, reported to contribute to diverse soil processes (47–50), which also corroborates with the detection of the bacteria in our analyses regardless of the ecotypes. We used ANOVA followed by Tukey *post hoc* test (51), and showed that the relative abundance of Proteobacteria (F = 7.292, *P* = 0.004) and Thaumarchaeota (F = 4.451, *P* = 0.020) differed between the dry and wet ecotypes. Proteobacteria were more abundant in the dry than in the wet ecotype, whereas Thaumarchaeota abundance was the opposite (Fig. 3). Some Proteobacteria may improve plant performance and growth, and can increase in abundance under drought conditions (52, 53), suggesting that Proteobacteria might be important for the sustainable growth of *A. gerardii* under the challenging environmental conditions in Colby. Thaumarchaeota are the dominant archaea in soil systems (54), and well-known ammonia oxidizers (55). We surmise that Thaumarchaeota in our study might have the potential to enhance the resilience of the *A.gerardii* wet ecotype under abiotic stressful conditions through the transformation of ammonia into nitrate (56).

Post-hoc SIMPER analyses at the Phylum level showed that Proteobacteria, Actinobacteria, Acidobacteria, Chloroflexi, Bacteroidetes and Verrucomicrobia contributed most to the differences among the dry, mesic and wet ecotypes (Table S2 in the supplemental material). Using *post hoc* SIMPER analyses, we observed that Actinobacteria, Acidobacteria, and Proteobacteria contributed to the greatest differences between the dry and mesic ecotypes, dry and wet ecotypes, and mesic and wet ecotypes. Verrucomicrobia and Bacteroidetes contributed to the greatest differences

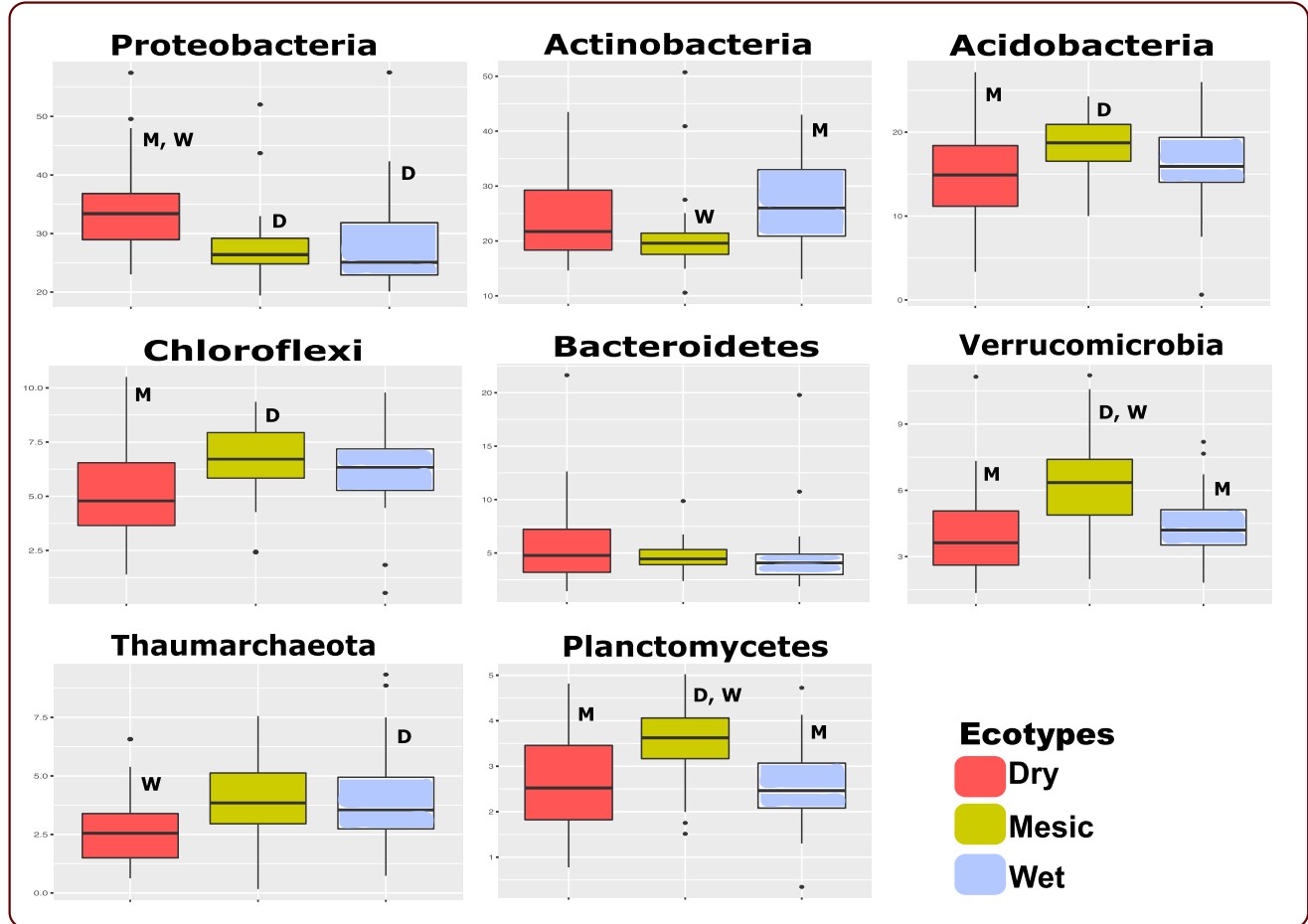

**FIG 3** The relative abundance of the top seven bacterial and one archaeal taxa present in all the three ecotypes. Proteobacteria and Thaumarchaeota were significantly different between the dry and wet ecotypes. Letters in a box plot are significantly different at $P < 0.05$ (D, M, W = significantly different from dry, mesic, and wet ecotypes, respectively).

between the dry and mesic ecotypes (Table S2). Planctomycetes and Thaumarchaeota contributed to the differences between all the ecotypes (Table S2). Comparing the dry and wet ecotypes, we observed that Bacteroidetes, and Thaumarchaeota contributed to their greatest differences. Verrucomicrobia, and Thaumarchaeota contributed most to the differences between the mesic and wet ecotypes. Putting it all together, despite the different ecotypic rhizobiome sharing 98.66% bacterial and archaeal ASVs, the relative abundances of the bacterial populations in the different ecotypes resulted in the ecotypic bacterial compositional differences (Table S2).

We used DeSEQ2 ($P < 0.05$, Fig. 4) and detected that *Rhizobium* had significant differences between the dry and wet ecotypic rhizobiomes. *Rhizobium* had higher relative abundances in the dry ecotype. Comparing dry and mesic ecotypes, we noticed that *Rhizobium*, *Pseudomonas*, *Cellulomonas*, *Rhodococcus*, *Parviterribacter*, *Parasegetibacter*, *Flavihumibacter*, *Cellvibrio*, and *Candidatus Berkiella* were more dominant in dry ecotype than in mesic ecotype. On the opposite side, *Microbispora*, *Sorangium*, *Zavarzinella*, and *Candidatus Udaeobacter* had more dominance in mesic than in wet ecotype. In other studies, *Rhizobium* has been found to be drought-stress tolerant (57), and well-known to aid plants during drought conditions (58). Putting it all together, our study suggested that *Rhizobium*, being the most predominant in the dry ecotype, might have the potential influence to benefit the host in the dry environments. This may also help to explain the higher leaf nitrogen concentrations and higher chlorophyll absorbance we observed in the dry ecotype, regardless of planting location (36, 38). We compared the differences between mesic and wet ecotypes as well, and observed that *Parafrigoribacterium*,

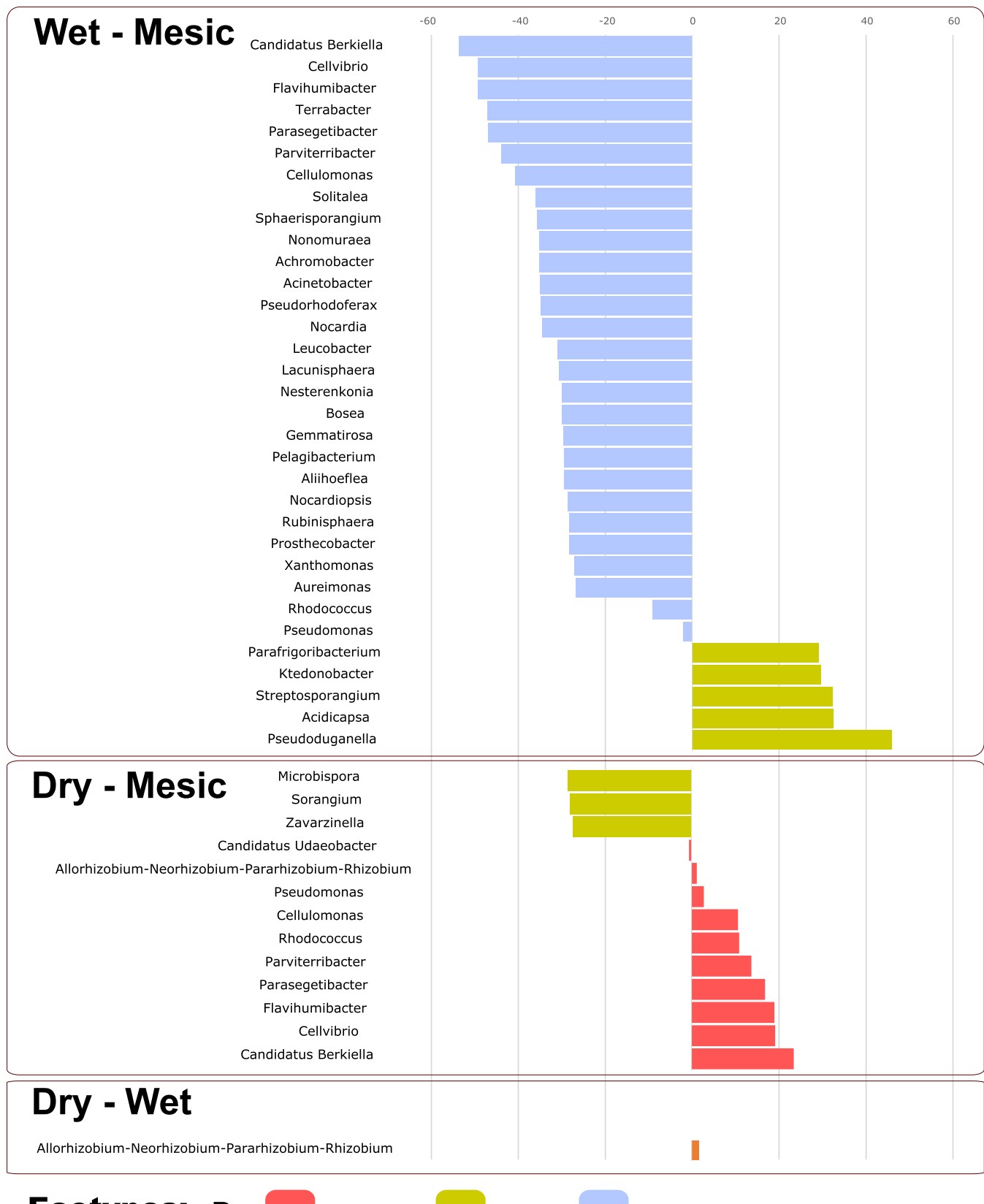

**FIG 4** DeSEQ2 analysis to reveal the bacterial genera that were significantly different in relative abundance between the dry, mesic, and wet ecotypes ($P < 0.05$). Pairwise comparisons were performed between wet-mesic, dry-mesic, and dry-wet ecotypes.

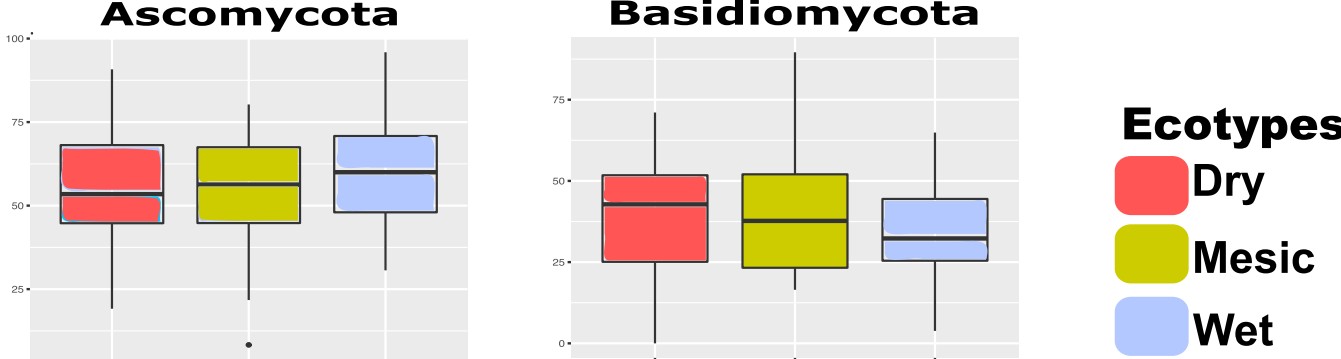

**FIG 5** The relative abundance of Ascomycota and Basidiomycota in the dry, mesic and wet ecotypes. Ascomycota and Basidiomycota are also the most abundant phyla in all the ecotypes.

*Ktedonobacter, Streptosporangium, Acidicapsa*, and *Pseudoduganella* were more dominant in the mesic ecotype. On the contrary, the top genera that were more predominant in wet ecotypes compared to the mesic belong to *Candidatus Berkiella, Cellvibrio, Flavihumibacter, Terrabacter, Parasegetibacter, Parviterribacter, Cellulomonas, Solitalea, Sphaerisporangium, Nonomuraea, Achromobacter, Acinetobacter, Pseudorhodoferax, Nocardia, Leucobacter*, among others (Table S3 in the supplemental material). *Leucobacter* has been identified to grow in wet, low-light environments (59), and we observed that *Leucobacter* had higher relative abundance in the wet ecotype (60) suggesting that *Leucobacter* might be better adapted to the wet environment.

**No differences in ecotypic fungi composition.** Unlike the observed differences in rhizobiome bacterial composition among ecotypes, we did not notice the same pattern in the ecotypic fungi composition. We observed that the ecotypic fungal compositions were not significantly different from each other at the phylum level when we performed the PERMANOVA statistical analyses (PERMANOVA: Pseudo-F = 2.0827, p(perm) = 0.071, p(MC) = 0.08). Pairwise Kruskal-Wallis tests also indicated no significant differences in the rhizosphere fungal community composition between the ecotypes (dry v wet: p(MC) = 0.190, dry v mesic: p(MC) = 0.117, wet v mesic: p(MC) = 0.072). The top eight fungal phyla present in all three ecotypes were Ascomycota, Basidiomycota, Mortierellomycota, Mucoromycota, Glomeromycota, Chytridiomycota, Kickxellomycota, and Rozellomycota (Fig. 5, Fig. S2 in the supplemental material). Besides that Ascomycota and Basidiomycota had the highest relative abundance, *post hoc* SIMPER analyses at the phylum level also indicated that Ascomycota and Basidiomycota were the top phyla contributing to the similarities in all the three ecotypes (Fig. 5, Table S2). Consistent with the results shown in our study, Ascomycota dominate the rhizosphere fungal communities (61–63). Basidiomycota has been reported to be isolated from soil as well, and is well-known for rhizobiome dwellers (64, 65). At the genus level, *Phallus* and *Cladosporium* (SIMPER analysis) were among the top genera contributing to the similarities between the ecotypes (Table S2).

**Soil carbon and nitrogen ratio differences between the ecotypes.** We performed soil physicochemical properties and measured soil %Carbon (%C) and %Nitrogen (%N) (Table S4 in the supplemental material). There was no significant difference between ratio of %Carbon and %Nitrogen (C/N) of dry and mesic rhizospheric soil (ANOVA: F = 0.802, $P$ = 0.373). However, we observed significant (at $P < 0.1$ level) differences between soil C/N of dry v wet (ANOVA: F = 3.032, $P$ = 0.085), and mesic v wet ecotypes (ANOVA: F = 5.323, $P$ = 0.024) (Fig. 6). We surmised that while the plant host and its associated rhizobiome had an influence on the soil biochemistry (66), more insights could be gained based on the microbial function in addition to the taxonomic identity. The huge diversity in the rhizosphere is known to perform multiple microbial functions, with high functional redundancy among the microbial members (67).

**Ecotype differences in bacterial diversity evident under arid conditions.** The distribution of microorganisms in an environment is often expressed in the famous

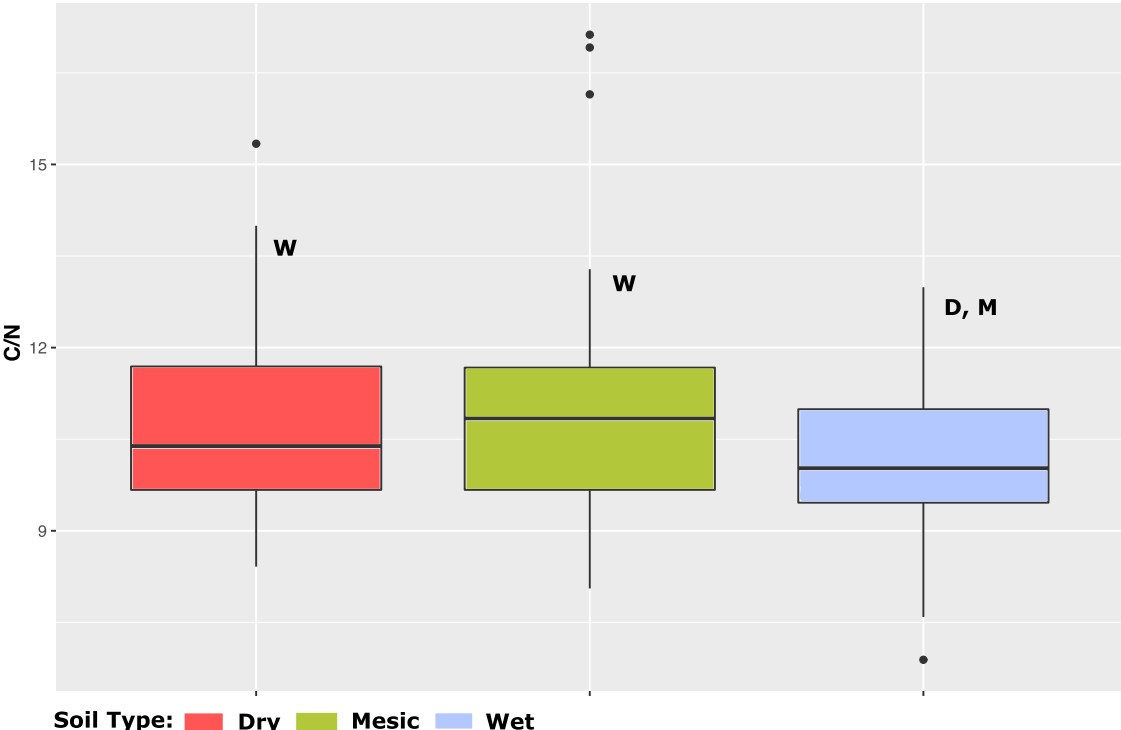

**FIG 6** Soil % carbon and % nitrogen ratio (C/N) between the ecotypes. Letters in a box plot are significant different at $P < 0.1$. (D, M, and W, significantly different from dry, mesic, and wet soil types, respectively).

tenet "everything is everywhere but the environment selects" (68). Our 10-year-long reciprocal common garden study provided an excellent opportunity to gain insights into whether and how specifically adapted ecotypic microorganisms will thrive and proliferate under arid conditions. The most ideal situation to understand these differences in bacterial diversity would include the ability to compare and contrast this study with the rhizobiome profiles before planting the *A. gerardii* ecotypes in Colby. Although this study is limited by the lack of time-series samples throughout the last 10 years, we deduced that after 10 years of planting and maintaining under drought stress conditions, the environmental pressure would alter the rhizobiome composition, especially when the hosts or ecotypes had limited or no influences (69–72). On the contrary, we observed that there were differences in bacterial community composition in perennial grassland ecotypes under drought stress, although there were no effects on the fungal compositions. Even without the before planting rhizobiome profiles, the bacterial community composition between ecotypes did in fact differ in the arid environment, suggesting that the host-mediated adaptation persisted even under arid conditions. The earlier report associated with *A. gerardii* has indicated that the ecotypic variation was observable in several aspects of growth in the plant. The ecotypic variation not only impacted aboveground features but also had a prominent effect on the belowground ecosystem processes mediated by microbial communities (37). We observed the higher relative abundance of *Rhizobium* in the dry ecotypic rhizosphere, suggesting that the *Rhizobium* might be associated with the observed higher leaf nitrogen concentrations and chlorophyll absorbance (36, 38). So, synthesizing results from our study and information from previous publications, we provided more insights to potential microbial populations in our study that might help to help the *A. gerardii* ecotypes to be more resilient to drought stress. The plant host can implement diverse mechanisms by secreting root exudates, employing defense strategies and structural modifications to recruit a specific and optimized microbiome (73). We demonstrated in our study that the individual plant genotype might influence the bacterial rhizobiome, and other reports showed that these beneficial plant genotypic traits capable of impacting the

rhizobiome could also be heritable (35, 74–76). There is a fine margin between the influence by the ecotype host and the environment on the microbiome communities. In our present work, we were able to determine that after 10 years of growing under conditions outside the normal ecotypic environment, the host ecotype might be able to "overcome" environmental pressure to a certain extent and regulate the bacterial community composition. Previous studies have indicated that drought stress can escalate the relative abundance of fungal populations while decreasing the relative abundance of bacterial communities in the same poplar plantation (77). Also, it is known that bacterial networks in soil are less stable than the fungal networks (78). Putting all these together, our study suggested that the ecotypic fungal populations might be more capable at adapting, and could have diverse resistance mechanisms toward drought conditions, resulting in the fungal community composition remaining unchanged. Moving forward, future time series association between plant physiological, genotypic and associated microbial community analyses would provide further insights into the impact of plant-microbe interaction.

In this study, we explored the concept of microbial "generalists" and "specialists". Generalist microbial populations are able to adapt to diverse habitats, while microbial populations are referred to as specialists when those can only adapt to specific habitats (79). There are previous reports which acknowledge the contribution of the generalists and specialists impacting the dynamics of the different microbial communities (80, 81). We observed that the differences in bacterial community composition were more prominent between dry and mesic as well as wet and mesic, compared to dry and wet ecotypes. While we were not able to identify specific bacterial populations as generalists and specialists from this study, we postulated that dry and wet ecotypic bacterial populations could be driven by specialists, living on the wet and arid margin of the plant species range. On the other hand, bacterial populations from the intermediate mesic ecotype might be guided more by the generalists. There are clear challenges in identifying specialist and generalist bacterial and fungal populations; understanding the co-existences of the specialists and generalists; and roles these specialists and generalists played in enhancing plant host resistance during environmental stress. Future work in elucidating the identity and functions of plant associated microbial specialists and generalists will provide insights in addressing these challenges. From this study, we surmised that *A. gerardii* ecotypes had unique bacterial assemblage contributing to the rhizobiome. However, the classification of whether the bacterial populations were generalists or specialists might have an influence on the resultant ecotypic rhizobiome due to host and environmental interaction.

To conclude, our study provided the knowledge that will help tackle the challenges faced by grassland restorative efforts by providing insights into the impact of environmental stress on plant host-associated microbiomes. We showed that bacterial populations were influenced by the respective ecotypes, while the fungal populations were not significantly different between the ecotypes. This study also suggested the existence of host-mediated bacterial community adaptation for *A.gerardii's* rhizosphere, and a possible tradeoff between the specialist and generalist bacterial communities in specific environments, that might ultimately benefit the plant host. The plant microbiome and its derived functions can substantially extend the plant hosts' adaptive capacity and resilience to a variety of environmental stressors (82). In the Great Plains, droughts are common (83) and productivity is limited (84), especially in tallgrass prairies. This study provides novel insights into the understanding of the impact of the rhizobiome on the enhancement of drought tolerance of *A. gerardii*, and its implications in grassland restoration efforts and management.

## MATERIALS AND METHODS

**Experimental design and sampling.** The common garden in Colby is located at the Kansas State University Agricultural Research Center in Thomas County (39°23′N, 101°04′W). The common garden was established in 2010-10 years before our sampling. The seeds of four populations of each of native dry (Hays), mesic (Manhattan, KS), and wet (Carbondale Illinois) *A. gerardii* ecotypes (36) were

germinated and then grown inside the greenhouse in potting mix substrate (Metro-Mix 510). Established 3-4-month-old plants from all the populations were then planted in western Kansas (Colby) as a surrogate for the drier conditions expected in the future (85). The size of the Colby common garden plot was 67.5 m². Each ecotype was represented by four populations with 12 replicate plants (36). There was a total of 12 plants (4 populations × 3 ecotypes) in a randomized complete block design with 10 blocks (36). Plants were planted 0.75 m apart along each row, and the soil around the plants was covered with a water-penetrable cloth to control unwanted plants. Some plants did not survive through the 10 years in the common garden. We collected a total of 95 rhizosphere soil cores (15 cm deep, 1.25 cm diameter) from the dry ($n = 33$), mesic ($n = 30$), and wet ($n = 32$) ecotypes during the growing season in Summer 2019. We considered the topsoil (0-15cm) to assess the impacts of grass ecotypes on rhizobiome. There are previous reports of using the topsoil to analyze the microbiome composition and diversity since the topsoil is considered to contain the most diverse microorganisms (86, 87). Each soil core was placed in a plastic bag, transported on ice, and stored at −80°C until genomic DNA extraction.

**Soil total nitrogen and carbon analyses.** We passed each soil sample through a 4-mm sieve to homogenize soil and remove large roots, and handpicked small roots from each soil sample for 10 min per sample. A 15 g subsample of sieved soil was then dried at 55°C for 1 week, grounded to a fine powder in a mixer mill (SPEX Instruments, Metuchen, NJ), and re-dried at 55°C. Approximately 50 mg ground soil was analyzed for %C and %N using dry combustion followed by gas chromatography on a Thermo Scientific FlashSmart 2000 NC Soil Analyzer (Milan, Italy). We used Analysis of Variance (ANOVA) in R to detect the statistical significance in %Carbon and %Nitrogen ratio (C/N) between the dry, mesic, and wet ecotypes (51).

**DNA extraction, metabarcoding, and analyses.** We extracted the genomic DNA from the root samples and soil associated with it (0.150g each) using the Omega E.Z.N.A. Soil DNA Kit (Omega Bio-Tek, Inc., Norcross, GA, USA), with a slightly modified protocol. Bulk soil was separated from the rhizosphere soil by handshaking the roots gently, and any soil that was attached to the root was considered part of the rhizosphere. Briefly, we modified the protocol using a Qiagen TissueLyser II (Qiagen, Hilden, Germany) for 2 min at 20 rev/s, and eluted the purified DNA with a final volume of 100 $\mu$L. The extracted microbial DNA were sequenced on the Illumina MiSeq, with the 16S rRNA V4 region amplified using the primers 515F and 806R with barcodes (88), and the internal transcribed spacer region ITS2 amplified using the primers fITS7 (89–91) and ITS4 (92), at the Kansas State University Integrated Genomics Facility.

We acquired a total of 1,729,418 bacterial and 3,570,536 fungal sequencing reads before quality check (QC) and trimming (Table S1 in the supplemental material). We used QIIME 2 (v. 2019.7) to process the sequence data and to profile the rhizobiome communities (93). We used QIIME 2 plugin cutadapt (94) to remove the primer sequences; reads with no primer were discarded. Additionally, we used DADA2 (95) for quality control with the same parameters across different runs and truncated the reads to length where the 25th percentile of the reads had a quality score below 15. The pre-trained classifier offered by QIIME 2, using SILVA database (v. 132) was used for taxonomic assignment for bacteria. Similarly, the UNITE classifier was trained on the full reference "develop" sequences (version 8.2, release date 2020-2-20) (96) using QIIME 2 2020.2 before taxonomic assignment of the fungal reads. We rarefied the data set to normalize the differences in sequencing depth between the samples before estimating diversity indices (97). $\alpha$-diversity was calculated to present the species diversity in each sample (98). We estimated observed richness ($S_{Obs}$), Shannon's diversity (H'), and Faith's PD using a rarefied data set (8,023 reads for 16S; 10,258 reads for ITS). Observed richness ($S_{Obs}$) is defined as the species numbers observed in a sample/set of samples (99). Similar to the $\alpha$-diversity analysis, we used Bray Curtis distances to compare the compositional dissimilarity among the different ecotypes and used non-metric multidimensional scaling (NMDS) to visualize the distance matrices. Differences in the relative abundances of bacterial and fungal phyla among the ecotypes were analyzed using PERMANOVA in R followed by Tukey's *post hoc* test ($P < 0.05$) (51). SIMPER *post hoc* analyses were performed to identify those community members that contributed to the highest differences among the ecotypes (100). The cutoff used for all SIMPER analyses was 70% to list only the taxonomic groups that contributed highest to the similarity or differences among the ecotypes. We used DeSEQ2 and highlighted marked differences in the disproportionate relative abundance of bacteria taxa among the ecotypes ($P < 0.05$) (101).

**Data availability.** All raw sequence data is available in the NCBI under BioProject accession no. PRJNA772708 and biosamples SAMN22405120 to SAMN22405309. Additional information can be found in the supplementary sections, and the bacterial and fungal taxon assignments along with their counts in each sample are available in figshare https://doi.org/10.6084/m9.figshare.19469846.

## SUPPLEMENTAL MATERIAL

Supplemental material is available online only.
**SUPPLEMENTAL FILE 1**, PDF file, 1 MB.

## ACKNOWLEDGMENTS

The study is based upon the work supported by the National Science Foundation EPSCoR Award No. OIA-1656006 and matching support from the State of Kansas Board of Regents. This study was supported by the United States Department of Agriculture, National Institute of Food and Agriculture (USDA NIFA), under the Award Number:

2020-67019-31803. We are thankful to Alina Akhunova of Kansas State Integrated Genomics Facility for the help with 16S and ITS amplicon sequencing.

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
