## [Reviewer comments · Microbiology Spectrum]

Microbiology Spectrum

Bacterial but not fungal rhizosphere community composition differ among perennial grass ecotypes under abiotic environmental stress

Soumyadev Sarkar, Abigail Kamke, Kaitlyn Ward, Aoesta Rudick, Sara Baer, Qinghong Ran, Brandi Feehan, Shiva Thapa, Lauren Anderson, Matthew Galliard, Ari Jumpponen, Loretta Johnson, and Sonny Lee

Corresponding Author(s): Sonny Lee, Kansas State University

Review Timeline:

Submission Date:	November 30, 2021
Editorial Decision:	February 4, 2022
Revision Received:	March 31, 2022
Accepted:	April 1, 2022

Editor: Kristen DeAngelis

Reviewer(s): Disclosure of reviewer identity is with reference to reviewer comments included in decision letter(s). The following individuals involved in review of your submission have agreed to reveal their identity: Ghada Dawwam (Reviewer #1); Xin Jing (Reviewer #2)

Transaction Report:

DOI: <https://doi.org/10.1128/spectrum.02391-21>

February 4, 2022

Dr. Sonny T.M. Lee
Kansas State University
Division of Biology
Manhattan

Re: Spectrum02391-21 (Bacterial but not fungal rhizosphere communities differ among perennial grass ecotypes under abiotic environmental stress)

Dear Dr. Sonny T.M. Lee:

Thank you for submitting your manuscript to Microbiology Spectrum. Two scientists reviewed your article, and both thought your study was interesting and well written in general. One reviewer was concerned that your conclusion is not well supported by the findings of this manuscript. I hope you can address this in your revised manuscript.

Link Not Available

Sincerely,

Kristen DeAngelis

Journals Department
Reviewer comments:

Reviewer #1 (Comments for the Author):

Dear Dr., Kristen DeAngelis
Editor, Microbiology Spectrum

The paper entitled " Bacterial but not fungal rhizosphere communities differ among perennial grass ecotypes under abiotic environmental stress" represents a good study of the impact of drought on the rhizobiome, and how the rhizobiome may modulate host performance and ameliorate its response to drought stress. The authors analyzed bacterial and fungal communities in the rhizospheres of three ecotypes (dry, mesic, and wet) of a dominant prairie grass, *Andropogon gerardii*. Some issues need clarification please.

Comments to the Author

1- The authors analyzed bacterial and fungal communities in the rhizospheres of three ecotypes (dry, mesic, and wet) of a dominant prairie grass, *Andropogon gerardii* and the ecotypes were established in 2010 under persistent dry conditions at the arid margin of the species' range in Colby Kansas.

Why didn't you take rhizosphere samples at different times (eg., zero time, after 5 years and after 10 years at different

conditions?. If they did that, they could be able to compare microbiome changes for the same condition plus different conditions.
2- This study will aid to optimize plant productivity through the use of different ecotypes under future abiotic environmental stress, especially drought.

I think if you performed just greenhouse experiment by using different ecotypes that you have determined in your paper and applied to plant under drought stress. I am sure it would be better.

3- Line 56, 57: please add the reference of this article "Drought Stress Triggers Shifts in the Root Microbial Community and Alters Functional Categories in the Microbial Gene Pool".

4- Did the authors make different soil analysis?

As chemical and physical analysis for soils under different conditions are very important.

5- Line 192, host ecotype ecotypic rhizobiome? What is the meaning?

Reviewer #2 (Comments for the Author):

This manuscript focuses on how grass ecotypes influence soil microbiomes under drought stress. Sarkar and colleagues used a common garden experiment and investigated the establishment of three ecotypes of Big Bluestem and their impacts on soil microbial and fungal community diversity and composition. They found that bacterial communities, but not fungal communities, are different in composition among the dry, mesic, and wet ecotypes of Big Bluestem. This work presents a timely and important topic on climate change and its potential influences on plant-soil interactions, with a new perspective of belowground microbiomes. This work has implications for grassland restoration, management, and protection.

I have two major concerns as follows:

1) I understand that there are 98.66% bacterial and archaeal ASVs that are common among the three ecotypic rhizobiomes (L171-173), but I don't understand what caused the compositional differences (e.g., L187, L196). One way in my mind is to decompose the total beta-diversity into quantitative forms and the presence-absence forms of replacement and richness difference (<https://rdrr.io/cran/adespatial/man/beta.div.comp.html>). So that you can show the major drivers of changes in bacterial composition.

2) I like the concept of microbial "generalists" and "specialists" (L310), but I am not convinced that "the bacterial population from the dry and wet ecotypes might be specialists and are less adaptive to the induced environmental stress" (L311-312). First, it is unclear which bacterial population belongs to the specialists. Second, there are groups of fungi that can also have specialized relationships with the host plants, such as pathotrophic and symbiotrophic fungi. Finally, I did not find firsthand evidence that the bacterial population from the dry and wet ecotypes are less adaptive to environmental stress.

Minor concerns

L1: If I understand correctly, the differences of bacterial communities are community composition, not alpha-diversity. If so, for clarity, it would be nice to replace communities with community composition.

L20-21: I understand that there is a need to understand "how the rhizobiome may modulate host performance and ameliorate its response to drought stress", but I don't understand how it relates to the research question of this study.

L33-34: Can you provide more specific information in the Discussion about the implications of this work? In particular, the information would be relevant to how to optimize plant productivity through the use of different ecotypes regarding the rhizobiome.

L56-57: I agree. There is a need to fill the knowledge gaps.

L62-64: The following sentence "Climate change ..." just shifts the topic, but it would be nice if you can provide more information to support the point that "Dissecting the rhizosphere bacterial and fungal communities ... are crucial to ..." before shifting the topic.

L67: I did not find any support information on the ecotypes of the dominant species above. Can you define the ecotypes of the dominant species?

L99-101: Can you present some working hypotheses based on the postulation?

L107: What is the size of the common garden? I was wondering whether the size of soil cores (L118) is suitable for examining the impacts of grass ecotypes on rhizobiome.

L112: This could be updated to the sixth IPCC assessment report.

L139: Replace Silva with SILVA (refer to L163).

L142: Double check the reference "community 2019".

L144: Can you define the observed richness (Sobs)? In the following text, e.g., Figure 1, it was observed OTUs index, while in the main text, amplicon sequence variants (ASVs) were defined.

L145: Can you justify why use 8,023 reads for bacteria and 10,258 reads for fungi?

L146: I may have missed the results related to UniFrac, but can you clearly present the results regarding the UniFrac?

L152: Can you justify why select 70% as the cut-off?

L170: Can you define H? I was a bit confused what the statistic you used.

L188: Can you justify why use the PERMANOVA at the phylum level, rather than the genus or species level?

L189: Text is missing in the statistical section about the PERMANOVA.

L189: What do p (perm) and p (MC) mean?

L196: I was confused about the use of ecotypic bacterial diversity because you have reported the results of alpha-diversity above, but what ecotypic bacterial diversity does indicate. In addition, I was also confused about the use of structure, because diversity and composition are both components of structure, and you already reported the results above. It would be nice if you can reword the terms to specifically indicate what you want to define.

L208: Can you replace : with = after p or replace = with :?

L219: Again, why phylum level?

L254: See my comment on L196.

L298: Unclear. What are the traits?

Staff Comments:

Preparing Revision Guidelines

Please return the manuscript within 60 days; if you cannot complete the modification within this time period, please contact me. If you do not wish to modify the manuscript and prefer to submit it to another journal, please notify me of your decision immediately so that the manuscript may be formally withdrawn from consideration by Microbiology Spectrum.

Dear Dr., Kristen DeAngelis
Editor, Microbiology Spectrum

The paper entitled " Bacterial but not fungal rhizosphere communities differ among perennial grass ecotypes under abiotic environmental stress" represents a good study of the impact of drought on the rhizobiome, and how the rhizobiome may modulate host performance and ameliorate its response to drought stress. The authors analyzed bacterial and fungal communities in the rhizospheres of three ecotypes (dry, mesic, and wet) of a dominant prairie grass, *Andropogon gerardii*.

Some issues need clarification please.

Comments to the Author

1- The authors analyzed bacterial and fungal communities in the rhizospheres of three ecotypes (dry, mesic, and wet) of a dominant prairie grass, *Andropogon gerardii* and the ecotypes were established in 2010 under persistent dry conditions at the arid margin of the species' range in Colby Kansas.

Why didn't you take rhizosphere samples at different times (eg., zero time, after 5 years and after 10 years at different conditions?. If they did that, they could be able to compare microbiome changes for the same condition plus different conditions.

2- This study will aid to optimize plant productivity through the use of different ecotypes under future abiotic environmental stress, especially drought.

I think if you performed just greenhouse experiment by using different ecotypes that you have determined in your paper and applied to plant under drought stress. Iam sure it would be better.

3- Line 56, 57: please add the reference of this article "Drought Stress Triggers Shifts in the Root Microbial Community and Alters Functional Categories in the Microbial Gene Pool".

4- Did the authors make different soil analysis?

As chemical and physical analysis for soils under different conditions are very important.

5- Line 192, host ecotype ecotypic rhizobiome? What is the meaning?

Dear Dr., Kristen DeAngelis
Editor, Microbiology Spectrum

We are thankful to the reviewers for their suggestions and comments. We have made changes to the manuscript (highlighted in blue) based on their comments. Briefly, we

- Adjusted our language to ensure clarity of our data analyses as well as ensuring our discussions and conclusions reflect our data.
- We also added the soil biochemistry total carbon and total nitrogen analyses to provide more insights into our study.

We have included a revised “clean” and “marked up” version for your perusal. The line numbers mentioned here correspond to the source text file of the “marked up” version. We hope that you and the reviewers will agree that changes to our manuscript provide more clarity of our work to the readers of Microbiology Spectrum.

Reviewer #1 (Comments for the Author):

Dear Dr., Kristen DeAngelis
Editor, Microbiology Spectrum

The paper entitled " Bacterial but not fungal rhizosphere communities differ among perennial grass ecotypes under abiotic environmental stress" represents a good study of the impact of drought on the rhizobiome, and how the rhizobiome may modulate host performance and ameliorate its response to drought stress. The authors analyzed bacterial and fungal communities in the rhizospheres of three ecotypes (dry, mesic, and wet) of a dominant prairie grass, *Andropogon gerardii*.

Some issues need clarification please.

Comments to the Author

1- The authors analyzed bacterial and fungal communities in the rhizospheres of three ecotypes (dry, mesic, and wet) of a dominant prairie grass, *Andropogon gerardii* and the ecotypes were established in 2010 under persistent dry conditions at the arid margin of the species' range in Colby Kansas.

Why didn't you take rhizosphere samples at different times (eg., zero time, after 5 years and after 10 years at different conditions?. If they did that, they could be able to compare microbiome changes for the same condition plus different conditions.

We agree with the reviewer that it would provide tremendous insights into shifts in the rhizobiome if we were able to observe and compare microbiome changes across time points (0 yrs, 5 yrs, 10yrs). Unfortunately, samples were not available prior to this present collection, but we feel that this present study would still provide crucial information on the effects of environmental and plant ecotypes on the rhizobiome to the scientific community. We understand the lack of past information is the limitation of this study, and we have mentioned this in the manuscript (Lines 322-325). Moving ahead, we are planning to collect samples every year, and looking forward to providing even more comprehensive insights into the interactive impact of the environment and host on the microbiome (Lines 353-355).

2- This study will aid to optimize plant productivity through the use of different ecotypes under future abiotic environmental stress, especially drought.

I think if you performed just greenhouse experiment by using different ecotypes that you have determined in your paper and applied to plant under drought stress. I am sure it would be better. Thank you for this suggestion. We really appreciate the comment that a greenhouse experiment would have been a nice approach to this study. Although we performed the existing study in the common garden plots, we plan to perform the greenhouse experiment in the future.

3- Line 56, 57: please add the reference of this article "Drought Stress Triggers Shifts in the Root Microbial Community and Alters Functional Categories in the Microbial Gene Pool".

We have added the reference (Lines 66, 432-434).

4- Did the authors make different soil analysis?

As chemical and physical analysis for soils under different conditions are very important.

We have performed additional soil analyses and measured the Soil's Total Carbon and Total Nitrogen parameters. We used the Total Carbon/Total Nitrogen ratio (C/N) to determine if there

were any significant differences in the C/N ratio among the ecotypic rhizosphere soil. The methods and results are presented on lines 142-150, 304-313.

5- Line 192, host ecotype ecotypic rhizobiome? What is the meaning?

We have deleted the word “ecotype” (Line 216).

Reviewer #2 (Comments for the Author):

This manuscript focuses on how grass ecotypes influence soil microbiomes under drought stress. Sarkar and colleagues used a common garden experiment and investigated the establishment of three ecotypes of Big Bluestem and their impacts on soil microbial and fungal community diversity and composition. They found that bacterial communities, but not fungal communities, are different in composition among the dry, mesic, and wet ecotypes of Big Bluestem. This work presents a timely and important topic on climate change and its potential influences on plant-soil interactions, with a new perspective of belowground microbiomes. This work has implications for grassland restoration, management, and protection.

I have two major concerns as follows:

1) I understand that there are 98.66% bacterial and archaeal ASVs that are common among the three ecotypic rhizobiomes (L171-173), but I don't understand what caused the compositional differences (e.g., L187, L196). One way in my mind is to decompose the total beta-diversity into quantitative forms and the presence-absence forms of replacement and richness difference (<https://rdr.io/cran/adespatial/man/beta.div.comp.html>). So that you can show the major drivers of changes in bacterial composition.

We thank the reviewer for this valuable comment. We used the post-hoc SIMPER analysis to identify the phyla that contributed to the differences between the ecotypes. In addition, we also included the mean and standard error values for the ASVs that were common among the three ecotypic rhizobiomes (Supplementary Table S2). Taken together with the post-hoc SIMPER analyses and the relative abundance of the common ASVs, we are confident that it will ensure

clarity for the readers. The concept and interpretation have been added to the manuscript (Lines 259-262).

Detailed explanations of the mean and standard errors, and SIMPER analysis follows:

The post-hoc SIMPER analysis measures the average percent contribution of the individual phylum to the dissimilarity between ecotypes in a Bray-Curtis dissimilarity matrix. This allowed us to identify the phyla that are major contributors to any differences between the ecotypes. In this study, we observed that the Actinobacteria, Proteobacteria, Acidobacteria, Verrucomicrobia, Bacteroidetes, Thaumarchaeota, Chloroflexi, Firmicutes, Patescibacteria, Planctomycetes, Armatimonadetes, Gemmatimonadetes, Latescibacteria, Cyanobacteria, Rokubacteria, Entotheonellaeota, Nitrospirae, BRC1, Chlamydiae, Dependuntiae, FBP, Elusimicrobia, Deinococcus-Thermus, Fibrobacteres, and WS2 are shared among the ecotypes. We used SIMPER analysis, and identified Proteobacteria [comparison between which ecotypes: percentage contribution to the differences] (dry and mesic: 8.69%, dry and wet: 9.81%, mesic and wet: 7.85%), Actinobacteria (dry and mesic: 9.23%, dry and wet: 10.06%, mesic and wet: 11.81%), Acidobacteria (dry and mesic: 9.01%, dry and wet: 9.45%, mesic and wet: 7.81%), Chloroflexi (dry and mesic: 5.98%, dry and wet: 5.9%, mesic and wet: 5.07%), Bacteroidetes (dry and mesic: 6.86%, dry and wet: 7.98%, mesic and wet: 5.74%) and Verrucomicrobia (dry and mesic: 7.6%, dry and wet: 5.56%, mesic and wet: 7%), Planctomycetes (dry and mesic: 4.53%, dry and wet: 4.18%, mesic and wet: 4.6%), and Thaumarchaeota (dry and mesic: 6.73%, dry and wet: 6.88%, mesic and wet: 6.76%) contributed most to the differences between the dry, mesic, and wet ecotypes. (Supplementary Table S2).

We have calculated the mean and standard errors of the relative abundance of the dominant phyla in the respective ecotypes (Supplementary Table S2). The phyla that are shared among the ecotypes are Actinobacteria (Dry: 0.24 ± 0.01 , Mesic: 0.20 ± 0.01 , Wet: 0.27 ± 0.01), Proteobacteria (Dry: 0.34 ± 0.01 ; Mesic: 0.28 ± 0.01 , Wet: 0.28 ± 0.01), Acidobacteria (Dry: 0.14 ± 0.01 , Mesic: 0.18 ± 0.006 , Wet: 0.16 ± 0.008), Verrucomicrobia (Dry: 0.04 ± 0.003 , Mesic: 0.06 ± 0.004 , Wet: 0.04 ± 0.002), Bacteroidetes (Dry: 0.05 ± 0.007 , Mesic: 0.04 ± 0.002 , Wet: 0.04 ± 0.005), Thaumarchaeota (Dry: 0.02 ± 0.002 , Mesic: 0.03 ± 0.003 , Wet: 0.04 ± 0.003), Chloroflexi (Dry: 0.05 ± 0.003 , Mesic: 0.06 ± 0.003 , Wet: 0.06 ± 0.003), Firmicutes

(Dry: 0.01 ± 0.002 , Mesic: 0.02 ± 0.002 , Wet: 0.02 ± 0.001), Patescibacteria (Dry: 0.01 ± 0.001 , Mesic: 0.02 ± 0.003 , Wet: 0.01 ± 0.001), Planctomycetes (Dry: 0.02 ± 0.001 , Mesic: 0.03 ± 0.001 , Wet: 0.02 ± 0.001), Armatimonadetes (Dry: 0.006 ± 0.0007 , Mesic: 0.009 ± 0.0008 , Wet: 0.006 ± 0.0004), Gemmatimonadetes (Dry: 0.008 ± 0.0007 , Mesic: 0.011 ± 0.0006 , Wet: 0.01 ± 0.0007), Latescibacteria (Dry: 0.001 ± 0.0002 , Mesic: 0.002 ± 0.0002 , Wet: 0.001 ± 0.0002), Cyanobacteria (Dry: 0.0009 ± 0.0003 , Mesic: 0.001 ± 0.0008 , Wet: $0.0003 \pm 9.44907E-05$), Rokubacteria (Dry: 0.0018 ± 0.0002 , Mesic: 0.001 ± 0.0002 , Wet: 0.002 ± 0.0002), Enttheonellaeota (Dry: 0.002 ± 0.0002 , Mesic: 0.002 ± 0.0002 , Wet: 0.002 ± 0.0002), Nitrospirae (Dry: 0.0012 ± 0.0001 , Mesic: 0.001 ± 0.0002 , Wet: 0.001 ± 0.0002), BRC1 (Dry: $0.0007 \pm 9.48591E-05$, Mesic: 0.001 ± 0.0001 , Wet: 0.0009 ± 0.0001), Chlamydiae (Dry: $0.0003 \pm 7.21611E-05$, Mesic: $0.0003 \pm 5.91075E-05$, Wet: $0.0003 \pm 6.89405E-05$), Dependitiae (Dry: $0.0001 \pm 3.86587E-05$, Mesic: $0.0002 \pm 5.51051E-05$, Wet: $0.0002 \pm 5.7477E-05$), FBP (Dry: $0.0002 \pm 4.3776E-05$, Mesic: $0.0001 \pm 3.5129E-05$, Wet: $0.0003 \pm 9.16193E-05$), Elusimicrobia (Dry: $9.7294E-05 \pm 3.39706E-05$, Mesic: $0.0002 \pm 5.14341E-05$, Wet: $0.0001 \pm 6.2104E-05$), Deinococcus-Thermus (Dry: $0.0001 \pm 4.0167E-05$, Mesic: $0.0001 \pm 3.64633E-05$, Wet: $8.55256E-05 \pm 2.172E-05$), Fibrobacteres (Dry: $4.91564E-05 \pm 2.36833E-05$, Mesic: $0.0001 \pm 3.27272E-05$, Wet: $0.000104807 \pm 3.22797E-05$), and WS2 (Dry: $1.42201E-05 \pm 1.42201E-05$, Mesic: $5.41088E-05 \pm 3.14157E-05$, Wet: $3.72345E-05 \pm 2.142E-05$).

The mean and the standard error of the relative abundance for the respective phylum suggested that although the phyla are shared between the ecotypes, the relative abundances differed. This can result in compositional differences although there is a high sharing percentage for Amplicon Sequence Variants (ASVs).

2) I like the concept of microbial "generalists" and "specialists" (L310), but I am not convinced that "the bacterial population from the dry and wet ecotypes might be specialists and are less adaptive to the induced environmental stress" (L311-312). First, it is unclear which bacterial population belongs to the specialists. Second, there are groups of fungi that can also have specialized relationships with the host plants, such as pathotrophic and symbiotrophic fungi. Finally, I did not find firsthand evidence that the bacterial population from the dry and wet ecotypes are less adaptive to environmental stress.

We are glad that the reviewer likes our discussion about the concept of microbial “generalists” and “specialists”, while we are also aware of the limitations of our argument. In order to improve the clarity of our manuscript, we have deleted the statement and introduced the downside of this claim based on the reviewer's comments, and modified the manuscript accordingly (Lines 366-371).

We have also introduced the following in the manuscript to clearly reflect the results of our study (Lines 362-366):

While we were not able to identify specific bacterial populations as generalists and specialists from this study, we postulated that dry and wet ecotypic bacterial populations could be driven by specialists, living on the wet and arid margin of the plant species range. On the other hand, bacterial populations from the intermediate mesic ecotype might be guided more by the generalists.

Minor concerns

L1: If I understand correctly, the differences of bacterial communities are community composition, not alpha-diversity. If so, for clarity, it would be nice to replace communities with community composition.

The manuscript title has been modified. “communities” has been replaced with “community composition” throughout the manuscript.

L20-21: I understand that there is a need to understand "how the rhizobiome may modulate host performance and ameliorate its response to drought stress", but I don't understand how it relates to the research question of this study.

We thank the reviewer for this comment. The research question of this study is whether and to what extent do the different ecotypes maintain or recruit distinct rhizobiomes after ten years in an arid climate. We hypothesized that we would identify microbial populations in the rhizobiome that might be more resilient to the drought stress as well as contribute to the plant host performance. Synthesizing results from our study and information from previous publications, we provided more insights to potential microbial populations in our study that might help to help the *Andropogon gerardii* ecotypes to be more resilient to drought stress (Lines 335-338).

We also have the following information in the manuscript:

(1) Proteobacteria were more abundant in the dry than in the wet ecotype, whereas Thaumarchaeota abundance was the opposite (Figure 3). Some Proteobacteria may improve plant performance and growth and can increase in abundance under drought conditions (Jang et al. 2020; Kim et al. 2011), suggesting that Proteobacteria might be important for the sustainable growth of *A.gerardii* under the challenging environmental conditions in Colby. Thaumarchaeota are the dominant archaea in soil systems (Schleper and Nicol 2010), and well-known ammonia oxidizers (Stieglmeier, Alves, and Schleper 2014). We surmise that Thaumarchaeota in our study might have the potential to enhance the resilience of the *A.gerardii* wet ecotype under abiotic stressful conditions through the transformation of ammonia into nitrate (Taffner et al. 2018) (Lines 241-248).

(2) In other studies, *Rhizobium* has been found to be drought-stress tolerant (Rehman and Nautiyal 2002), and well-known to aid plants during drought conditions (Staudinger et al. 2016). Putting it all together, our study suggested that *Rhizobium*, being the most predominant in the dry ecotype, might have the potential influence to benefit the host in the dry environments. This may also help to explain the higher leaf nitrogen concentrations and higher chlorophyll absorbance we observed in the dry ecotype, regardless of planting location (Caudle et al.2014, Galliard et al. 2020) (Lines 269-274).

L33-34: Can you provide more specific information in the Discussion about the implications of this work? In particular, the information would be relevant to how to optimize plant productivity through the use of different ecotypes regarding the rhizobiome.

Specific information has been provided in the results and discussion section (Lines 381-387).

L56-57: I agree. There is a need to fill the knowledge gaps.

We are glad that the reviewer shared our sentiment on the need to fill these crucial knowledge gaps.

L62-64: The following sentence "Climate change ..." just shifts the topic, but it would nice if you can provide more information to support the point that "Dissecting the rhizosphere bacterial and fungal communities ... are crucial to ..." before shifting the topic.

We have added more information regarding "Dissecting the rhizosphere bacterial and fungal communities..." to improve the readability of the manuscript (Lines 72-80).

L67: I did not find any support information on the ecotypes of the dominant species above. Can you define the ecotypes of the dominant species?

We have rephrased the statement (Lines 82-84).

L99-101: Can you present some working hypotheses based on the postulation?

We have stated the working hypothesis (Lines 114-119).

L107: What is the size of the common garden? I was wondering whether the size of soil cores (L118) is suitable for examining the impacts of grass ecotypes on rhizobiome.

The size of the common garden plot is 67.5 m². The plot has rows with plants 0.75m apart, 12 plants in a row in a randomized complete block design with 10 blocks (Lines 129-133).

We considered the topsoil (0-15cm) to assess the impacts of grass ecotypes on rhizobiome. There are previous reports of using the topsoil to analyze the microbiome composition and diversity since the topsoil is considered to contain the most diverse microorganisms (Hao Jingjie et al.; Wright et al. 2022). We have added this information to the revised manuscript (Lines 136-138)

L112: This could be updated to the sixth IPCC assessment report.

The reference has been updated (Lines 129, 535-538).

L139: Replace Silva with SILVA (refer to L163).

We have replaced Silva with SILVA (Lines 169).

L142: Double check the reference "community 2019".

We have double-checked the reference. The complete citation is:

Community, Unite. 2019. “UNITE QIIME Release for Fungi.” *UNITE Community*.

L144: Can you define the observed richness (S_{obs})? In the following text, e.g., Figure 1, it was observed OTUs index, while in the main text, amplicon sequence variants (ASVs) were defined. Observed richness (S_{obs}) is defined as the species numbers observed in a sample/set of samples (Lines 176-177). We have replaced the observed OTUs index with the observed amplicon sequence variants (ASVs) index throughout the manuscript and the figure for clarity.

L145: Can you justify why use 8,023 reads for bacteria and 10,258 reads for fungi?

We have used a rarefied dataset to estimate the observed richness (Kleine Bardenhorst et al. 2022) (Lines 172-173). Rarefaction is the method that adjusts for differences in the library sizes across samples to ensure unbiased comparisons of alpha diversity. Rarefaction selects a definite number of samples that is either equal to or less than the number of samples in the smallest sample. Then the reads are randomly discarded from larger samples until the number of remaining samples is equal to this threshold. We obtained the number of reads for bacteria (8,023 reads) and fungi (10,258 reads) after performing the rarefaction.

L146: I may have missed the results related to UniFrac, but can you clearly present the results regarding the UniFrac?

We have mistakenly introduced the word “UniFrac”. In the revised manuscript, we have removed the word “UniFrac”. We have used Bray Curtis distances to compare the community dissimilarity among the different ecotypes.

L152: Can you justify why select 70% as the cut-off?

We selected Similarity of Percentage Analysis (SIMPER) with a 70% cut-off percentage to list only the taxonomic groups that were higher-contributing (Lines 183-185).

L170: Can you define H? I was a bit confused what the statistic you used.

We thank the reviewer for raising this concern. H value corresponds to the test statistic for the Kruskal-Wallis test. The distribution of chi-square approximates the H distribution. We performed the Kruskal-Wallis statistical analysis for differences in bacterial α -diversity among

the *Andropogon gerardii* ecotypes. The explanation has been added to the revised manuscript (Lines 197-199, 206-209).

L188: Can you justify why use the PERMANOVA at the phylum level, rather than the genus or species level?

We had performed the PERMANOVA analyses at the phylum as well as at the genus level. We used PERMANOVA to identify if there was a statistical significance at a higher taxonomic level (phylum) before we performed additional statistical analyses on the lower taxonomic level (genus). This would ensure that we were not misled by false-negative results at lower taxonomic levels (genus) due to more repeated tests. Furthermore, this enabled us to use differential analyses (DeSEQ2) to detect the differential abundance of the bacterial genera between different ecotypes. We have added the PERMANOVA statistics for the genus level in the revised manuscript (Lines 219-221). Genus: (PERMANOVA: Pseudo-F 2.1014, p ([permutation (perm)]: 0.001, p [Monte Carlo (MC)]: 0.003).

L189: Text is missing in the statistical section about the PERMANOVA

We added the missing text (Lines 217-219).

L189: What do p (perm) and p (MC) mean?

p(perm) is the permutation P-values: P-values obtained using permutations.

p(MC) is the Monte Carlo P-values: Corrected P-values may also be obtained through Monte Carlo random draws from the asymptotic permutation distribution in the event that too few permutations are available for a given test.

We have included both p(perm) and p(MC) values to ensure that readers are aware of the statistical differences between the ecotypic rhizobiomes all both levels. We have mentioned the meanings of p(perm) and p(MC) in the manuscript (Lines 217-219).

L196: I was confused about the use of ecotypic bacterial diversity because you have reported the results of alpha-diversity above, but what ecotypic bacterial diversity does indicate. In addition, I was also confused about the use of structure, because diversity and composition are both

components of structure, and you already reported the results above. It would be nice if you can reword the terms to specifically indicate what you want to define.

The word “structure” has been removed. We have reworded the term ecotypic bacterial diversity to bacterial composition. We have used the term diversity to refer to alpha-diversity. This change has been made throughout the manuscript.

L208: Can you replace : with = after p or replace = with :?

We have replaced: with = (Lines 239-240) and ensure consistency in formatting throughout the revised manuscript.

L219: Again, why phylum level?

We used post-hoc SIMPER to identify the phyla that were the major drivers of changes in bacterial composition between the ecotypes (phylum) before we performed additional post-hoc SIMPER analyses on the lower taxonomic level (genus). This would ensure that we are not misled by false-negative results at lower taxonomic levels (genus) where there are more repeated tests. So, instead of doing the post-hoc SIMPER at the genus level, we performed differential analyses (DeSEQ2) to detect the differential abundance of the bacterial genera between different ecotypes. We reported SIMPER analysis at the phylum and genus level for fungi. Also, *Phallus* and *Cladosporium* were among the top fungal genera contributing to the similarities between the ecotypes (Supplementary Table S2).

L254: See my comment on L196.

We have removed the word “structure” and reworded the term ecotypic fungi diversity and structure to fungi composition (Lines 285).

L298: Unclear. What are the traits?

We have added the following to improve the clarity of the revised manuscript: (Lines 340-342).

We demonstrated in our study that the individual plant genotype might influence the bacterial rhizobiome, and other reports showed that these beneficial traits of plant genotypes capable of

impacting the microbiome associated with the roots could also be heritable (Schweitzer et al. 2008; Peiffer et al. 2013; Walters et al. 2018; Gray et al. 2014).

April 1, 2022

Dr. Sonny T.M. Lee
Kansas State University
Division of Biology
Manhattan

Re: Spectrum02391-21R1 (Bacterial but not fungal rhizosphere community composition differ among perennial grass ecotypes under abiotic environmental stress)

Dear Dr. Sonny T.M. Lee:

Thank you for your careful attention to revisions for this article. Your manuscript has been accepted, and I am forwarding it to the ASM Journals Department for publication. You will be notified when your proofs are ready to be viewed.

Sincerely,

Kristen DeAngelis
Editor, Microbiology Spectrum

Journals Department
Supplemental Material FOR Publication: Accept